# Cultural Influence on Corporate Sustainability: A Board of Directors Perspective

Diana Escandon-Barbosa *, Jairo Salas-Paramo and José Luis Duque

Facultad De Ciencias Económicas y Administrativas, Pontificia Universidad Javeriana Cali, Cali 760031, Colombia; jasalas@javerianacali.edu.co (J.S.-P.); jose.duque@javerianacali.edu.co (J.L.D.)
* Correspondence: dmescandon@javerianacali.edu.co

**Abstract:** This research aims to analyze the triple moderating effect of the board of directors in the country culture of a firm and its influence on the relationship between organizational innovation and organizational learning in corporate sustainability. A survey of 400 exporting companies of different commercial products from Colombia, Peru, Ecuador, and Bolivia was used to carry out this research. We used the structural equations model to explore the analysis of the causal and moderation relationships between the variables under study. As a result, it was found that the influence of the board of directors of a firm is essential for innovation processes because they drive their results to corporate sustainability. This last approach is due to the strategic approach adopted by large companies. In the case of SMEs, it was not possible to demonstrate that the board of directors has such a degree of influence. In the case of the moderating effect of the board of directors on the country's culture, it was possible to observe that the board of directors becomes a factor in the firm's performance despite its geographical location, which determines the influence of culture on its operation in corporations such as SMEs.

**Keywords:** board of directors; country culture; organizational innovation; organizational learning; corporate sustainability; multinational firms; SMEs





## 1. Introduction

In the ever-evolving realm of corporate governance, the role of boards of directors has transformed from a passive overseer to an active architect of an organization's strategic vision and sustainability agenda. The literature in the field of board directors, especially in international business, has been characterized by a focus on corporate governance (Chatjuthamard et al. 2023; Bolton and Zhao 2022; De Beule et al. 2022; Raimo et al. 2022; Asni and Agustia 2022; Attia et al. 2021), corporate social responsibility (Shelton et al. 2022; Ma and Chen 2023; Pucheta-Martínez et al. 2022), corporate culture (Azcorra et al. 2023; Pi and Yang 2023), CEO characteristics (Heubeck and Meckl 2023; Aksoy et al. 2023), countries' studies (Lozano and Martínez-Ferrero 2022), and sustainability (Zhao et al. 2022).

This study delves into the intricate dynamics between board leadership, organizational innovation, organizational learning, and the broader sphere of sustainability within the context of manufacturing enterprises across Colombia, Peru, Ecuador, and Bolivia. Against the backdrop of a shifting corporate governance landscape, marked by shareholder activism, evolving legal and regulatory norms (Johnson et al. 2019), and disruptive technological innovations, boards have emerged as pivotal agents in shaping the future of organizations (Hermalin and Weisbach 2017). The global business community grapples with profound environmental and societal challenges, and boards are increasingly tasked with navigating these complexities while steering organizations toward sustainable practices.

Drawing inspiration from institutional theory (DiMaggio and Powell 1983), which offers insight into how firms respond to environmental challenges and institutional norms, this research sets out to explore and validate the intricate relationships between board

dynamics and sustainability innovation performance. This study contributes to the field by empirically confirming the links between organizational innovation, organizational learning, board characteristics, country culture, and sustainability outcomes.

A key focus of this study is the cultural context's role in moderating organizational innovation's impact on sustainability performance (Gelfand et al. 2007). We uncover how cultural nuances can influence the effectiveness of board-led sustainability initiatives, shedding light on the importance of cultural adaptation in pursuing sustainability goals. From an empirical point of view, it is possible to observe great interest in knowing the influence of the board of directors not only on the performance of the firm, but also concerning the decision processes, where significant and negative relationships have been found regarding its control (Li et al. 2022).

Our findings hold practical significance for both practitioners and scholars. Organizations can leverage these insights to foster innovative cultures, allocate resources to research and development (Davenport and Prusak 1998), and tailor their strategies to diverse cultural contexts. Additionally, we underscore the importance of board characteristics, including size (Hermalin and Weisbach 2017), in conjunction with organizational learning and innovation, offering guidance for optimizing governance structures to enhance sustainability performance.

This research invites scholars and practitioners in corporate governance to explore the intricate relationships between governance, innovation, culture, and sustainability. It provides a roadmap for leveraging board dynamics to catalyze sustainable and innovative corporate futures. In doing so, we aim to contribute to the evolving narrative of corporate governance, fostering a more sustainable and prosperous global business landscape.

To fulfill the purpose of this study, the paper has been structured as follows. In the first part, an analysis of the board of directors theory is carried out, analyzing the foundational elements of the field, as well as the primary studies in the area. The second part presents the study's conceptual framework that supports the paper's main postulates. In the third part, a methodological approach is carried out, proposing the selected study model, the processing of the information, and the results obtained. In the fourth part, the main conclusions are presented, as well as the main contributions to the area of study and the managerial field.

## 2. Theoretical Framework

### 2.1. Institutional Theory and Board Directors

Wong et al. (2023) used the institutional theory in the field of board directors as a theoretical framework that allows explaining environmental factors that may contribute to the understanding of stakeholders. Another important aspect is that it highlights the relevance of the firm's environment, especially the regulatory pressures that legitimize corporate practice (DiMaggio and Powell 1983). An essential aspect of this approach is that the institutional theory highlights that firms tend to develop homogeneous practices over time; similarly, they are exposed to environmental pressures that influence their habitual behaviors.

On the other hand, these practices tend to develop a legitimacy that tends to be concentrated in three ways: coercive, mimetic, and normative (DiMaggio and Powell 1983). In the case of coercion, it results from the interaction between formal and informal pressures that induce the organization to certain types of behavior. In the case of the mimetic, it is part of the decision of the organizations to adapt practices that are recognized as the best in the sector. Finally, the normative alludes to the pressures generated in the environment, increasing the laws determining the firm's aspirations for human talent.

It is essential to highlight that the institutional theory's primary value focuses on how the firm's structures and processes respond to institutional pressures (DiMaggio and Powell 1983). This phenomenon is called institutionalization, when the firms and their board directors are assimilated in behavior, structures, and processes. This last phenomenon allows the characterization of practices related to structures and their comparability. In the

same way, institutional theory allows the interpretation of norms, values, and beliefs that are legitimized and determine behaviors (Wong et al. 2023).

The relationship between institutional theory and the board of directors is in the importance of corporate governance practices, decision-making processes, and the organizational behavior developed by firms (De Villiers and Alexander 2014). From a sociological perspective, the contribution that institutional theory makes is that it provides the tools to examine how forms and individuals establish social norms, values, and institutional rules (Lewellyn and Muller-Kahle 2023). From the information above, it can be derived that organizations adopt certain practices and structures not only for efficiency, but also to achieve their objectives through legitimizing and accepting stakeholders, such as investors, customers, regulators, and society.

The direct influence of institutional theory on the board of directors can occur with the following elements. The first has to do with compliance with institutional norms. Usually, the board of directors maintains the norms and standards of corporate governance (Peng and Chandarasupsang 2023). Since they accept government practices, the rules of the sector and industry associations show adherence to the rules and ensure the legitimacy of the organization before the stakeholders. The second element is related to mimetic isomorphism, which establishes as a practice the imitation of successful practices recognized among peers (Hung et al. 2023). This imitation occurs to the extent that the board directors analyze the practices carried out by those who run recognized companies in the sector. This practice aims to legitimize the firm's position in the sector and develop successful management models.

The third element is regulatory pressures. In this sense, the direct boards face the pressures of the different stakeholders (Renz et al. 2023). These pressures can lead firms to adopt sustainable practices, including diversity and inclusion policies and social responsibility practices that align with societal expectations (Bhatia and Gulati 2023). The fourth element is related to cognitive isomorphism. In this sense, the board directors are exposed to the cognitive models and beliefs in the institutional environment of the firm (Mazza et al. 2023). This process allows board directors to share governance, ethics, and corporate responsibility factors. Finally, the last element is legitimation strategies. In this element, the board directors commit to developing strategies to demonstrate alignment between the firm and the values and norms institutionalized in the sector (Saggese et al. 2023). This condition may go hand in hand with the need to adapt to the pressures of the environment and share values with institutional pressures.

Finally, the last element that is also important to highlight is that the board directors' size is defined as a critical factor that influences the effectiveness of the board and its influence on the firm's performance (Martino et al. 2012). These seminal investigations have stated that from the beginning, the involvement of board directors in the strategy has been fundamental to understanding the behaviors of the different members and the strategic approaches adopted (Sahoo et al. 2023). In this way, it is possible to understand how the board of directors can improve organizational performance, especially in terms of its sustainability (Boshnak et al. 2023).

Scholars such as Berhe (2023) propose that the board of directors' structure is directly related to the firm's performance. Likewise, its board composition and the gender diversity integrated into its activities have a positive and significant relationship in sustainability, as well as in the independence of the board of directors. Despite this, a negative relationship was also found in a limited size of the board of directors, since a limited size creates conditions in which board directors begin to have limitations in the efficiency of decisions, consequently affecting the firm's results.

### 2.2. Adaptation to the Institutional Environment and Organizational Strategies

It is considered that institutional theory is established as a dynamic component that includes factors that influence the interaction between different actors (Athar et al. 2023). It is possible to observe that the relationship between organizational innovation, organi-

zational learning, the sustainability innovation performance, and the board directors is based on the behavior that firms assume, the government structure, and the strategies that ensure success in the long-term context in which they operate (Melis and Nawaz 2023; Jouber 2023).

Therefore, the institutional theory provides the lenses from which it is possible to understand how organizations and their board directors respond to the pressures and regulations of the context that regulate the search for innovation and sustainability (Naveed et al. 2023). In this way, institutional pressures and organizational innovation vary with the stakeholders' expectations (Athar et al. 2023). The board of directors acts by the institutional norms and the expectations related to the culture of innovation within the firm. In this way, the board directors assume the investment risks in research and development that may support strategic initiatives to promote innovative practices in the industry (Debellis et al. 2023).

On the other hand, in terms of institutional norms and organizational learning, it is established that the norms determine the degree of acceptance of the firm's practices. In this way, when the norms of the environment influence the board directors, they prioritize learning to adapt to changes in the environment successfully (Hudson and Morgan 2022). Learning from previous experiences, industry practices, and environmental trends become fundamental elements that seek to maintain the firm's sustainability (Nemoto 2022).

Regarding institutional legitimacy and sustainability innovation performance, organizations seek to legitimize themselves by complying with social expectations and norms, which include sustainable practices (Athar et al. 2023). The board directors will manage these expectations while being instrumental or valuable in formulating sustainable strategies to monitor performance concerning environmental, social, and governance environments (Nemoto 2022).

In the case of institutional isomorphism and governance practices, board directors are under the influence of isomorphism, which leads them to adopt governance practices of companies with a good reputation that are recognized for their corporate scope (Hussein et al. 2022). This mimicry leads them to carry out the diffusion of sustainable practices through the industry that helps their legitimacy in the environment in which they operate (Hu et al. 2022). Regarding legitimization and sustainable initiatives, the board directors use legitimization strategies to align the actions of companies in establishing institutional norms (Dobija et al. 2023).

Similarly, the interest in legitimizing their practices highlights environmental work and, especially, efforts for social responsibility. These efforts lead the organization to social approval and support (Di et al. 2022). Finally, the institutional theory guides understanding of the relationships between organizational innovation, organizational learning, sustainability innovation performance, board directors, and corporate governance practices (Meng et al. 2022). The preceding concepts demonstrate that the influence of regulations and organizational pressures exerted on the board of directors influence strategies presenting innovation and improvement of long-term sustainability (Nemoto 2022).

### 2.3. Board Directors

The relationship between the board directors and the organizational innovation, organizational learning, and sustainability innovation performance is a function of how the company determines the strategic direction, the ethical elements, and especially the long-term success (Pongelli et al. 2023). In the case of the board directors and their commitment to the firm's innovation, their vision will support the culture of innovation that empowers the firm's employees to think creatively, take calculated risks, and develop solutions to face the challenges of the environment (Hudson and Morgan 2023).

Likewise, the board directors recognize the importance of continuous learning and sharing knowledge within the organization through the prioritization of learning initiatives and the mental disposition to permanent learning. Concerning the firm's sustainable governance, the board of directors is responsible for verifying the company's performance,

including developing sustainable practices (Gorla et al. 2023). Another aspect is monitoring innovative and sustainable performance that assures the firm of its adherence to the dynamics of the environment and the stakeholders' expectations (Fuentes et al. 2023).

The board of directors becomes a central factor in the long-term vision, especially in the sustainability of the strategy (Gerged et al. 2023). When sustainability is a focus for the firm, it is disseminated within the firm in a comprehensive manner and as a commitment to long-term success (Bendig and Ernst 2022). This environment also allows the creation of a culture that allows the balance between innovation activities and associated risks through experimentation.

Gerged et al. (2023) affirm that the size of the board of directors is essential for the excellent operation of the company's strategies. In this regard, the size of the board of directors is established as an essential field of research on the firm's corporate governance (Padungsaksawasdi et al. 2021). Studies in the field have shown that prominent board directors tend to drive innovation processes more effectively, creating the conditions to create a strong culture (Gerged et al. 2023).

On the other hand, the size of the board of directors is also associated with the amount of resources to be managed. This dynamic demands knowledge, and skill is necessary to consider not only the complex dynamics of the execution of activities, but also articulation with the firm's innovation culture. Therefore, a large board of directors will allow greater access to the resources located in the external environment of the firm, especially regarding technological and financial factors necessary for innovation (Jackling and Johl 2009).

Other studies have also shown the relationship between innovation culture and board size. This relationship directly influences behavioral governance and indicates the critical success factors of innovative culture (Gerged et al. 2023). Likewise, the size of the board of directors is primarily associated with the establishment and execution of policies, strategies, and results that are evident in the firm's results (Wang and Ellinger 2011).

Finally, with the commitment assumed by the stakeholders, together with the transparency in their activities, the board directors reinforce the organization's commitment with a commitment to the environment and the activities of social responsibility (Cuevas-Rodríguez et al. 2023). The board of directors is a variable that influences the relationship between organizational innovation, organizational learning, sustainability innovation performance, and, especially, corporate governance (Shah and Ivascu 2023).

*2.4. Country Culture Role*

Ullah et al. (2022) argue that studying culture is essential since it allows us to understand and identify the different variables that influence countries in economic, political, and social development. Similarly, studies in the field suggest that the study of developing countries or, in the case of the current research, in emerging economies are highly influenced by regulations, technology, and some factors related to demand (Machokoto and Agyei-Boapeah 2018).

For the specific case of technology in developing or emerging countries, the country's economic development is an essential factor. The same is true for consumers since the demand for more specialized products increases over time (Sheth 2011). Much of the literature suggests a big difference in the factors determining growth in developed or emerging countries. These factors are decisive in the search for strategies that allow the execution of activities aimed at environmental innovation and the company's sustainability (Eckstein et al. 2018).

Based on the institutional theory perspective, culture is defined by informal institutions that are potential determinants of innovation and environmental sustainability (Ullah et al. 2022). From institutional theory, culture is also conceived as a normative pressure that determines the behavior of the actors through informal regulations. Some studies state that culture affects decision-making processes and corporate behavior (Peng and Zhang 2022). Finally, a country's culture will directly influence the actions to

improve the company's external environment, performance, and environmental practices (Wang et al. 2022).

*2.5. Organizational Innovation, Organizational Learning, and Sustainability Innovation Performance*

In companies' searches to find strategic alternatives to ensure their success in the long-term, firms choose to assume organizational innovation and learning as necessary alternatives to ensure a sustainable innovation performance in the long-term (Irfan et al. 2023; Xie et al. 2006). Regarding organizational innovation, the firm tries to introduce ideas, processes, and products that allow them to ensure an advantage in the market and, thus, sustained growth. On the other hand, the technological developments and the approaches used to solve problems primarily develop a culture that allows the company's employees to experiment and assume the risks necessary for the operation (Özgül and Zehir 2022).

Organizational learning addresses acquiring knowledge through experience in the firm's performance (Xie et al. 2006). This previous condition also goes hand in hand with shared knowledge and feedback. Concerning sustainability innovation performance, it takes advantage of the results in the implementation of innovation and learning strategies to improve growth and the implementation of responses to the dynamics of the environment (Jiménez-Jiménez and Sanz-Valle 2011).

Concerning organizational innovation as an organizational strategy, it allows firms not only to join efforts in the creation of products with high added value, but also creates the conditions for a culture of innovation that allows firms to explore new sustainable technologies, products, and practices (Cui et al. 2022). In the same way, organizational innovation promotes efficiency in resource use, especially in reducing waste and finding eco-friendly alternatives to protect the environment (Pandita et al. 2023; Wang and Ellinger 2011).

For scholars like Ober (2020), for companies to continue with their development process in the market, new ideas and solutions that are innovative are required. The danger of getting stuck and the need to improve its competitive position is achieved through organizational learning that allows improving the sustainability innovation performance of the company through the improvement of the knowledge and skill of the company's employees (Tuffour et al. 2023; Meeus et al. 2001). Based on the information above, the following hypothesis is proposed:

**Hypothesis 1:** *Organizational innovation positively and directly influences sustainability innovation performance.*

In the case of organizational learning as a strategy and organizational innovation, its objective is to support activities that improve sustainability innovation performance (Özgül and Zehir 2022). In the same way, organizational learning also makes it possible to identify the complexities and challenges of identifying opportunities for the improvement of the firm. In the same way, organizational learning allows the collection of information to analyze the firm's impact on the environment and society through the success and failures obtained.

Özgül and Zehir (2022) affirm that organizational learning becomes a knowledge-based process that becomes a source of competitive advantage. As a process of the firm, it allows the creation, acquisition, and transfer of knowledge to improve sustainability innovation performance (Migdadi 2020). On the other hand, for authors such as Makhloufi et al. (2021), organizational learning allows the firm to make use of its tangible and intangible resources that facilitate the development of capacities and, with them, practices for the improvement of the necessary performance for the sustainability innovation performance. Thus, the following hypothesis is posited:

**Hypothesis 2:** *Organizational learning positively and directly influences sustainability innovation performance.*

In the case of country culture, it is considered a central aspect of the innovation processes carried out by companies (Escandon et al. 2023). The cultural context of a country directly influences organizational innovation, which in turn directly impacts sustainability innovation performance. The preceding is based on the idea that a country's culture, organizational innovation, and sustainability innovation performance are multifaceted. According to Wang et al. (2023), innovation processes do not depend only on internal factors, but are also influenced by the context in which they operate.

In this way, culture, considered values, practices, and norms, can facilitate or restrict the adoption and integration of sustainable practices resulting from the firm's innovations (Escandon et al. 2023). Understanding the cultural context and its degree of importance is a critical position for firms in their search to improve their performance (Goma 2023). In the same way, by aligning innovation efforts with cultural values and country practices, conditions are created to generate more positive and effective results for better results for the different stakeholders of the firm (Hou et al. 2023). Considering the information above, the following hypothesis is posed:

**Hypothesis 3:** *Country culture moderates the relationship between organizational innovation and sustainability innovation performance.*

As in the case of organizational innovation, culture determines the dynamics of a context in which the organization generates learning processes based on its experience, and that allows it to have the capabilities to face the dynamics of the markets (Tyagi and Moses 2020; Xie 2019). In this sense, the relationship between country culture and organizational learning is based on the possibility of improving the organization's capacity to carry out sustainable innovations (Escandon et al. 2023). This type of practice is combined with the values, beliefs, and social norms that induce the adoption of practices that favor learning processes and consequently achieve the objectives related to sustainability through innovative practices (Abdul-Halim et al. 2019).

In this way, organizational learning is a process in which knowledge is acquired, shared, and applied, a practice influenced by culture (Escandon et al. 2023). For Hofstede, culture as a set of dimensions allows cross-functional collaboration and cooperation to prioritize knowledge for problem solutions. This idea is related to the influence of a country's culture, which can determine practices in sustainability innovation performance (Nawab et al. 2021). On the other hand, culture can also become a factor that determines resistance to change and risk aversion, where traditional practices prevent the adoption of approaches based on learning that can be directed towards sustainability (Othman and ElKady 2023).

Overcoming barriers is a crucial and necessary element to promote, especially when learning is required that allows it to be receptive to the needs of the context (Ju et al. 2021). This recognition of the profound influence of culture on learning and the subsequent effect on sustainability and innovation performance is a necessary framework to understand how cultural diversity, empowerment, and capacity development allow organizations to achieve their objectives for improvement through business performance, social welfare, and its contribution to sustainable development (Mousa et al. 2022).

**Hypothesis 4:** *Country culture moderates the relationship between organizational learning and sustainability innovation performance.*

The board of directors' size represents the decision makers of the firm that have direct influence within the organizations and plays an essential role in strengthening the culture of learning, especially in the orientation towards the sustainability approach for the firm (Matta et al. 2023; Ben Rejeb et al. 2020). In this way, the cultural context of a country conditions the approach assumed by the board directors in the face of efforts to develop activities for the firm's innovation, but significantly to improve learning processes that allow organizations to improve their levels of knowledge and experimentation (Behbahaninia 2022).

In turn, the board of directors' size influences the culture and how it is assumed within the firm, which allows for overcoming barriers to achieving better performance levels (Naheed et al. 2021).

In this way, board directors influence organizations' search to face the diversity of the global context, especially in the markets they serve (Cindrić 2021). This explains the importance of the board directors' size in considering culture as a determining factor and that they influence the firm through the different interactions within it (Louca et al. 2020). Firms can strengthen their commitment to sustainable development and contribute to better learning performance, especially in sustainability practices.

**Hypothesis 5:** *Country culture moderates the relationship between the board of directors' size and its interaction with organizational learning in predicting sustainability innovation performance.*

The board of directors, as well as the cultural context in which the firm operates, are determining factors for the development of skills that allow the use of learning and the achievement of objectives regarding the sustainability of innovations in performance (Ben Rejeb et al. 2020). As decision-makers, board directors play a role in constructing a culture for innovation since they are the central guides of the firm's strategies (Behbahaninia 2022). Furthermore, in the proper cultural context of a country, not only the focus of innovation and sustainability is determined, but also the ability to achieve results (Louca et al. 2020). The leadership and strategic vision developed by the board of directors' size directly influence organizational learning. In cultures where board directors focus their efforts on the search for innovation and sustainability, they begin to observe a strong propensity towards sustainability and innovation that support the firm's activities (Matta et al. 2023). In this way, the board of directors will also consider decision-making under risk, focusing on the firm's potential to achieve better performance.

It is in the culture itself that the board of directors' size allows for prioritizing and strengthening the social responsibility of the firm, as well as being aligned with the needs of the environment (Cindrić 2021). In the same way, culture allows adopting a culturally diverse approach and, at the same time, aligning strategies with norms and values that promote inclusive activities for the development of innovation capacities and ensure performance in maintaining sustainability in their innovation efforts (Shi et al. 2021; Moussa and Helfaya 2017). Considering the information above, the following hypothesis is posed:

**Hypothesis 6:** *Country culture moderates the relationship between the board of directors' size and its interaction with organizational innovation in predicting sustainability innovation performance.*

Conceptual model of hypothesis presented in Figure 1.

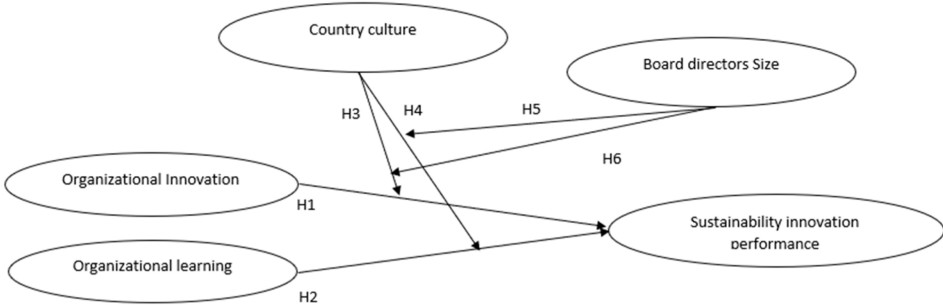

**Figure 1.** Conceptual Model.

## 3. Methodology

*3.1. Context*

In the dynamic landscape of contemporary business, understanding the intricate connections between culture, organizational innovation, and sustainability is of paramount

importance. This understanding gains added significance when contextualized within the specific framework of Colombia, Peru, Ecuador, and Bolivia. These four countries possess a distinct blend of economic, cultural, and regulatory influences that profoundly shape the behavior of organizations and their approach to sustainability.

These Latin American nations, while sharing a geographical region, are characterized by a rich tapestry of economic diversity. Colombia and Peru have emerged as relatively more prosperous economies, driven by a variety of sectors, including mining, agriculture, and services. Ecuador, on the other hand, exhibits economic fluctuations influenced by its reliance on oil exports. Bolivia, endowed with abundant natural resources, faces unique economic challenges rooted in historical factors. Understanding how these economic variations intersect with sustainability efforts is essential to crafting effective strategies.

Cultural heterogeneity is another hallmark of this region. Influenced by indigenous legacies, colonial histories, and contemporary globalization, these cultures play a pivotal role in shaping consumer behaviors, organizational practices, and perceptions of sustainability. Recognizing the nuances of these cultures is fundamental to devising sustainable innovation strategies that resonate with local contexts (Hofstede 1980).

Moreover, the regulatory landscape varies significantly across these countries. Colombia and Peru have adopted more pro-business policies, actively courting foreign investments. Meanwhile, Ecuador and Bolivia have taken a more interventionist stance, especially in sectors rich in natural resources. These regulatory divergences impact the incentives and limitations faced by businesses when adopting sustainable practices.

Given this contextual backdrop, this research endeavors to uncover the intricate interplay between culture, organizational innovation, and sustainability within these four Latin American nations. By doing so, it seeks to contribute to the broader discourse on corporate governance and sustainability, offering region-specific insights that can inform policy, strategy, and practice in these dynamic economies.

*3.2. Data*

A simple random sample of 1535 enterprises operating in Colombia, Peru, Ecuador, and Bolivia in 2022 was used to conduct this analysis. The businesses were chosen from databases of exporting businesses in all the nations in the customs registry of each country. Information was obtained from over 650 businesses in each country, but only 400 in Colombia, 380 in Peru, 395 in Ecuador, and 360 in Bolivia were considered.

Our research team collected data in Colombia to obtain comprehensive insight into the exporting business landscape (DANE 2022). The initial aim was to secure information from over 650 businesses to ensure a robust and representative sample. Ultimately, the final sample size reached 400 businesses, and was strategically designed to maintain representativeness while slightly below the initial target (DANE 2022). This sample allocation included 6.2% large corporations and 93.8% micro-, small-, and medium-sized enterprises (MIPYMES), following their distribution within the country's exporting sector (DANE 2022). This approach guarantees that the research findings reflect the diverse range of businesses operating in Colombia and their varying sizes.

Our research endeavors in Peru also aspired to gather insight from over 650 businesses, ensuring thorough representation of the exporting business population (INEI 2022). Ultimately, the sample size achieved was 380 businesses. This allocation, however, adhered to the same principles of proportionality, with 6.2% of the sample representing large corporations and 93.8% encompassing MIPYMES, mirroring the distribution observed within Peru's exporting landscape (INEI 2022). The meticulous sampling strategy ensures that our research offers valuable insight applicable to the rich tapestry of businesses operating in Peru.

Similarly, in Ecuador, where there are projected to be 10,050 exporting businesses in 2022, we aimed to include over 650 businesses in our sample (The Business Year: Ecuador 2022). The final sample comprised 395 businesses, with 11% being large corporations and 89% representing MIPYMES, aligning with their respective proportions in the exporting

sector (INEC 2022). In Bolivia, with 2344 exporting companies, the target sample size was also over 650, but the final sample consisted of 360 businesses. This allocation featured 4% large corporations and 96% MIPYMES, ensuring that our research accounts for the diversity in the Bolivian exporting business landscape (INE 2022). These meticulous sample size considerations affirm the representative nature of our research findings and their applicability to Ecuador and Bolivia's dynamic economic contexts.

### 3.3. Model Variables

The approach comprises three components: organizational innovation, organizational learning, and sustainable innovation performance.

Organizational innovation (OI): This scale was developed and tested by (Chen et al. 2020). This construct is measured in nine categories of an organization's ability to develop innovative activities. These things are rated on a scale of 1 to 7, where 1 is completely disagreed with, and 7 is wholly agreed with.

Organizational learning (OL): This scale is designed to measure organizational learning, specifically focusing on two dimensions of learning behavior within an organization: exploitative learning and exploratory learning (Atuahene-Gima and Murray 2007). Organizational learning refers to the process by which an organization acquires, retains, and utilizes knowledge to adapt to changing environments and improve performance.

Sustainability innovation performance (SIP): Drawing inspiration from Schöggl et al. (2020) and Ketata et al. (2014), the study identified four dimensions for the SIP: product sustainability, resource efficiency, environmental pollution, and social responsibility.

On the other hand, two variable dichotomies allow for managing fundamental relationships: the board of directors' size and country culture (BDSize). The size of the board of directors is measured on two levels: 0 if there are fewer than five active members and 1 if there are five or more members.

The choice of specifying the number "5" as a measurement base for a large board size was made considering the existing literature and standard industry practices. In numerous studies on corporate governance and board dynamics (Johnson et al. 2019), a threshold of five or more directors is frequently used to distinguish larger boards from smaller ones. This threshold is significant, as it often aligns with regulatory and governance recommendations, where a board size of five or above is associated with increased complexity and potential implications for corporate decision-making processes.

Finally, country culture (CC) is measured along two dimensions: performance orientation and human orientation. This division is based on the work of Inglehart and Baker (2000). The results of the Globe Project were used to classify each country participating in this study: performance orientation (high level of masculinity, uncertainty avoidance, power distance, and future orientation) and human orientation (high level of femininity, institutional, and societal collectivism).

Our approach incorporates control variables. As a result of the nature of our model, we incorporated variables such as size, sector, and years in international markets.

### 3.4. Model

The methodology employs a structural equation model (SEM) to simultaneously analyze the relationships among the variables and account for the moderating effects of board of directors' size and country culture.

The SEM consists of observed variables representing the items from the scales used to measure each variable and latent variables representing the constructs (organizational innovation, organizational learning, and sustainability innovation performance). The model includes interactions between the independent variables and the moderator variables, as well as interactions between the moderator variables themselves.

The SEM allows for a comprehensive analysis of the relationships and interactions between organizational innovation, organizational learning, and sustainability innovation

performance while considering the moderating effects of the board of directors' size and country culture.

The model has been estimated using appropriate software (e.g., Mplus) to obtain the relationships' parameter estimates, fit indices, and statistical significance. Fit indices, such as the chi-square test, comparative fit index (CFI), Tucker–Lewis index (TLI), root mean square error of approximation (RMSEA), and standardized root mean square residual (SRMR), will be used to assess the goodness-of-fit of the model.

By using SEM and considering the moderators, the study aims to provide a comprehensive understanding of how organizational innovation and organizational learning contribute to sustainability innovation performance and how these relationships are influenced by the size of the board of directors and the cultural context in each country. The findings from this analysis will contribute valuable insights into organizational innovation and sustainability.

### 3.5. Descriptive Statistics and Validity Measures

Table 1 presents the descriptive statistics, correlation coefficients, compound reliability, and average variance extracted for each measurement scale. The board director variable was not included in the table because it is a quantitative variable, but it was reported to have a mean of 4.5 members and a standard deviation of around 3.54.

**Table 1.** Mean, SD, CR, and AVE.

|   |     | Mean | SD    | 1    | 2    | 3    | 4 | CR   | AVE  |
|---|-----|------|-------|------|------|------|---|------|------|
| 1 | OI  | 4.11 | 0.819 | 1    |      |      |   | 0.91 | 0.68 |
| 2 | OL  | 4.35 | 0.761 | 0.11 | 1    |      |   | 0.86 | 0.77 |
| 3 | SIP | 4.67 | 1.134 | 0.43 | 0.31 | 1    |   | 0.88 | 0.74 |
| 4 | CC  | 5.12 | 1.056 | 0.23 | 0.27 | 0.32 | 1 | 0.81 | 0.72 |

Note: SD, standard deviation; CR, composite reliability; AVE, average variance extracted.

The compound reliability assesses the internal consistency of the scales, and the average variance extracted (AVE) measures the amount of variance explained by the construct relative to measurement error. To meet the criteria for convergent validity, the compound reliability should be higher than 0.7, and the AVE should be higher than 0.6 for each construct.

Additionally, the researchers confirmed discriminant validity by examining the confidence intervals determined for each construct pair. The absence of the value "1" in these intervals indicates that the constructs are distinct and not perfectly correlated with each other.

## 4. Results

In this section, we present the detailed findings of our study, which sought to investigate the relationships between organizational innovation, organizational learning, board size, country culture, and sustainability innovation performance in Colombian, Peruvian, Ecuadorian, and Bolivian manufacturing firms. We utilized a structural equation model (SEM) with interaction effects to thoroughly explore the hypothesized correlations and moderating influences.

Our findings robustly support Hypothesis 1. The relationship between organizational innovation and sustainability innovation performance was not only statistically significant ($\beta = 0.72$, $p < 0.001$) but also practically substantial. This result underscores the pivotal role of organizational innovation in directly and positively influencing sustainability outcomes. Manufacturing firms that actively engage in innovative methods outperform their peers in sustainability-related activities. Organizational innovation not only enhances products and processes, but also fosters a culture of sustainability, yielding positive environmental and social consequences.

While our findings support Hypothesis 2, indicating a significant positive association between organizational learning and sustainability innovation performance, it is noteworthy that the path coefficient is smaller in magnitude ($\beta = 0.33$, $p < 0.05$) compared to organizational innovation. This suggests that while organizational learning has a direct and positive impact on sustainability outcomes, it may not be as potent as organizational innovation in driving sustainability performance. Nonetheless, our findings underscore the critical importance of cultivating a learning-oriented organizational culture for improving sustainability practices and outcomes.

Our study reveals a significant moderating effect of country culture on the link between organizational innovation and sustainability innovation performance (H3) ($\beta = 0.45$, $p < 0.001$). This finding highlights the nuanced nature of the impact of organizational innovation on sustainability outcomes across nations with diverse cultural contexts. Further post hoc analyses indicate that countries with strong performance and future orientations experience a more pronounced beneficial influence of organizational innovation on sustainable performance. Conversely, countries characterized by a stronger human orientation and institutional collectivism demonstrate a less pronounced but still favorable association between organizational innovation and sustainability outcomes. These nuanced cultural variations emphasize the need for tailored approaches to organizational innovation for optimal sustainability performance across different regions.

Our data also support Hypothesis 4, indicating that country culture has a marginally significant moderating influence on the relationship between organizational learning and sustainability innovation performance ($\beta = 0.15$, $p < 0.05$). This suggests that each country's cultural background may alter the impact of organizational learning on sustainability outcomes to some extent. Further research is warranted to fully understand the cultural factors driving this interaction effect. This finding underscores the need for organizations to consider the cultural context when implementing learning-oriented initiatives to enhance sustainability practices.

Additionally, we conducted a comparative analysis to assess the impact of the board's size on our findings. Our results show that Hypothesis 5 related to country culture moderates the link between the board of directors' size and its interaction with organizational learning in predicting sustainability innovation performance ($\beta = 0.28$, $p < 0.001$). Importantly, this comparison highlights the influence of board size in conjunction with organizational learning on sustainability outcomes, with the cultural environment of each country playing a significant role. The t-value for this comparison was calculated as 2 ($p = 0.001$). This result indicates a statistically significant difference between firms with small and large board director sizes regarding the influence of board size on the interaction between organizational learning and sustainability innovation performance. Specifically, the moderating effect appears to be more pronounced and impactful in firms with larger board sizes.

This result underscores the influence of the cultural context on how board size and organizational learning jointly impact sustainability outcomes. Further analyses indicate that in countries with a high-performance and future orientation, the interaction between board size and organizational learning leads to more substantial improvements in sustainability performance. Conversely, countries characterized by a stronger human orientation and institutional collectivism demonstrate a less pronounced effect of this interaction on sustainability outcomes. When paired with robust organizational learning capacities, a larger board size resulted in more substantial improvements in sustainability performance in countries with high performance and future orientation. This result suggests that, in such contexts, a larger board can facilitate more effective learning processes that enhance sustainability outcomes.

Conversely, countries characterized by a stronger human orientation and institutional collectivism demonstrated a less apparent impact of the interplay between board size and organizational learning on sustainable results. Here, the cultural context mitigates the effects of board size and learning initiatives on sustainability performance.

Finally, our findings also strongly support Hypothesis 6, which posited that country culture moderates the relationship between the board of directors' size and its interaction with organizational innovation in predicting sustainability innovation performance. The data reveal a significant moderating effect ($\beta = 0.25$, $p < 0.001$). The t-value for this comparison was calculated as 2.45 ($p = 0.001$). This result indicates a statistically significant difference between firms with small and large board of directors sizes regarding the influence of board size on the interaction between organizational innovation and sustainability innovation performance. Specifically, the moderating effect appears more pronounced and impactful in firms with larger board sizes.

This result highlights the role of cultural context in shaping how board size and organizational innovation jointly influence sustainability outcomes. Further analyses indicate that in countries with a high performance and future orientation, the interaction between the board size and organizational innovation leads to more substantial improvements in sustainability performance. Conversely, countries characterized by a stronger human orientation and institutional collectivism demonstrate a less pronounced effect of this interaction on sustainability outcomes, as shown in Table 2.

**Table 2.** Results of model.

| Model Relation | Board Director Size | | | | |
| | Small | | Large | | |
| | Coefficient | *p* | Coefficient | *p* | *t* Test (Valor *p*) |
|---|---|---|---|---|---|
| OI | 0.72 | 0.00 *** | 0.75 | 0.000 | 0.65 (0.823) |
| OL | 0.33 | 0.005 ** | 0.41 | 0.05 | 1.92 (0.005) |
| Moderation OI * CC | 0.45 | 0.00 *** | 0.628 | 0.000 | 2.11 (0.000) |
| Moderation OL * CC | 0.15 | 0.005** | 0.175 | 0.05 | 2.45 (0.000) |

Note: * $p > 0.10$, ** $p < 0.05$, *** $p < 0.01$.

## 5. Conclusions

This study intended to explore the links between organizational innovation, organizational learning, board of directors' size, country culture, and sustainability innovation performance in manufacturing enterprises in Colombia, Peru, Ecuador, and Bolivia. Our research was supported by the theoretical framework of institutional theory, which provides valuable insight into how firms respond to environmental challenges and institutional norms, eventually influencing their sustainability practices (Wong et al. 2023; DiMaggio and Powell 1983).

Our research confirmed our predictions and provided helpful insight into the most significant variables' relationships. We found that organizational innovation had a positive and direct relationship with sustainable innovation performance, emphasizing the necessity of developing an innovative culture to promote sustainable outcomes (Irfan et al. 2023). Furthermore, country culture operated as a moderator, impacting the influence of organizational innovation on sustainability results. We discovered cultural disparities in how organizational innovation affects sustainability performance, with some nations seeing a more significant positive impact than others (Escandon et al. 2023).

In the case of organizational learning, it is confirmed that it has a significant relationship with sustainable innovation performance, indicating that to the extent that experience-based learning activities are developed, it allows the organization to better face the conditions and requirements of the environment, including management (Migdadi 2020). In this sense, the organization allows it to develop its knowledge transfer systems, and therefore, it is reflected in the results of sustainable innovation.

The study's findings have important implications for businesses to improve their sustainability performance. First, developing a creative culture and adopting innovative

practices can motivate sustainability projects (Abdul-Halim et al. 2019). Organizations should emphasize R&D investment, employee creativity, and cross-functional collaboration to create long-term innovation. Recognizing the moderating influence of country culture also demonstrates the importance of context-specific efforts to promote sustainability (Othman and ElKady 2023). Companies operating in different countries should understand the cultural factors influencing the relationship between innovation and sustainability to maximize their strategy (Mousa et al. 2022).

Additionally, our findings emphasize the significance of considering cultural characteristics when utilizing organizational learning to promote sustainability innovation performance (Matta et al. 2023). Organizations with international operations should adapt their strategy to account for cultural norms and expectations, providing confidence that their plans align with the local context to improve sustainability performance (Ben Rejeb et al. 2020). Therefore, adapting to the culture allows their tastes and preferences to be valued, generating better results in organizational learning and the requirements for creating environmentally sustainable products (Mousa et al. 2022).

On the other hand, our research highlights the importance of the board of directors' size in conjunction with organizational learning and organizational innovation in creating sustainable results (Matta et al. 2023). Understanding the cultural variables that govern these linkages enables firms to customize their tactics for supporting sustainable innovation in varied cultural contexts (Cindrić 2021). The board of directors' size was also highlighted, especially with regard to organizational learning and organizational innovation (Gerged et al. 2023). The cultural background of each country determined the combined influence of these elements on sustainability innovation performance (Louca et al. 2020).

We found that larger boards had a more substantial beneficial impact on organizational learning in countries with a strong culture (Jackling and Johl 2009). The relationship between organizational learning and sustainability innovation performance was reduced but still positive in nations with higher sociability cultures. We concluded that larger boards had a more substantial beneficial impact on organizational learning in countries with a strong performance and future orientation.

These results highlight the need for firms to modify their innovation strategies to meet the cultural context in which they operate. A large board of directors emphasizing organizational innovation may succeed in some cultural situations. Organizations can improve their capacity to promote sustainability innovation and connect their practices with the cultural expectations of their stakeholders by being aware of these differences (Cuevas-Rodríguez et al. 2023).

## 6. Scales

Sustainability Innovation Performance

This measures whether innovation enhances product sustainability, improves resource use efficiency, reduces environmental harm, and supports social responsibility initiatives, providing a comprehensive evaluation of the program's sustainability impact.

Our innovation activities have contributed to the success of our firm within the last three years with respect to:

*Dimension Product Design*

Use of low-impact materials;

Improved end-of-life phase of products (e.g., improve recyclability).

*Dimension Process Efficiency*

Efficient resource/material deployment;

Reduced resource consumption.

*Dimension Environmental Pollution*

Reduced environmental pollution and waste;

Decreased transportation and logistics.

*Dimension Social Responsibility*

Improved health and safety of employees;

Improved social and ethical situation.

Organizational Innovation

The scale covers diverse dimensions, including best practices, employee policies, quality management, collaboration, flexibility, and external partnerships, offering a comprehensive view of an organization's commitment to innovation.

Respond these aspects on a 1 to 7 scale, where 1 signifies strong disagreement and 7 indicates strong agreement:

1. The company I work for uses best practices and databases.

2. Emphasis is placed on employee development and retention practices in the company I work for.

3. The company I work for attaches importance to quality management systems.

4. Brainstorming is given importance in decision-making in my company.

5. Establishment of interdepartmental working groups is given importance in the company where I work.

6. The company I work for develops flexible working responsibilities.

7. The company I work for attaches importance to cooperation with customers.

8. The company I work for attaches importance to cooperation with suppliers.

9. The company I work for attaches importance to the use of outsourcing in its commercial activities.

Organizational Learning

Organizational learning refers to the process by which an organization acquires, retains, and utilizes knowledge to adapt to changing environments and improve its performance.

*Exploitative Learning* (1 = "strongly disagree," and 5 = "strongly agree")

Our aim was to search for information to refine common methods and ideas in solving problems in the project.

Our aim was to search for ideas and information that we can implement well to ensure productivity rather than those ideas that could lead to implementation mistakes in the project and in the marketplace.

We searched for the usual and generally proven methods and solutions to product development problems.

We used information acquisition methods (e.g., survey of current customers and competitors) that helped us understand and update the firm's current project and market experiences.

We emphasized the use of knowledge related to our existing project experience.

*Exploratory Learning* (1 = "strongly disagree," and 5 = "strongly agree")

In information search, we focused on acquiring knowledge of project strategies that involved experimentation and high market risks.

We preferred to collect information with no identifiable strategic market needs to ensure experimentation in the project.

**Author Contributions:** Conceptualization, D.E.-B.; software, D.E.-B.; validation, D.E.-B., J.S.-P. and J.L.D.; formal analysis, J.S.-P. and J.L.D.; investigation, D.E.-B., J.S.-P. and J.L.D.; data curation, D.E.-B.; writing—original draft preparation, D.E.-B., J.S.-P. and J.L.D.; writing—review and editing, D.E.-B., J.S.-P. and J.L.D.; visualization, D.E.-B., J.S.-P. and J.L.D.; supervision, D.E.-B., J.S.-P. and J.L.D.; project

administration, D.E.-B. and J.L.D.; All authors have read and agreed to the published version of the manuscript.

**Funding:** This research received no external funding.

**Informed Consent Statement:** Not applicable.

**Data Availability Statement:** The data used in this study was derived from a simple random sample of 1535 enterprises operating in Colombia, Peru, Ecuador, and Bolivia in the year 2022. These enterprises were selected from databases of exporting businesses registered with the customs authorities in each respective country. It is important to note that while information was initially obtained from over 650 businesses in each country to ensure a comprehensive and representative sample, only a subset was considered for the final analysis: 400 in Colombia, 380 in Peru, 395 in Ecuador, and 360 in Bolivia. Specifically, data pertaining to the Colombian enterprises was collected by our research team to provide a comprehensive understanding of the exporting business landscape. This information was gathered from official sources, including DANE (National Administrative Department of Statistics) in the year 2022. However, it is important to emphasize that due to the need for rigorous statistical analysis, the data availability statement in this study is not related to data accessibility but rather to the statistical significance of the results. For the statistical analyses conducted, the following significance levels were applied: * ($p > 0.10$), ** ($p < 0.05$), and *** ($p < 0.01$). These symbols denote the levels of statistical significance associated with the results of our analysis.

**Conflicts of Interest:** The authors declare no conflict of interest.

**Limitations and Future Research:** It is critical to recognize some of our study's limitations. First, the cross-sectional design limits our ability to prove causality definitively. Longitudinal approaches to study could be used in the future to obtain insight into dynamic relationships throughout time. Furthermore, our focus on manufacturing companies in certain South American countries may limit our findings' generalizability to other industries and locations. Extending the study to include a broader range of businesses and worldwide areas would expand the application of our findings.

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
