# Peer review of "Cultural Influence on Corporate Sustainability: A Board of Directors Perspective"

_ijfs, doi:10.3390/ijfs11040132_

Round 1

Reviewer 1 Report

Comments and Suggestions for Authors

·         In the abstract what do you mean by “For the information processing,”

·         The introduction section needs to be enhanced by adding a paragraph on the research contribution.

·         The authors need to justify the research context, why it is important to understand the subject matter within Colombia, Peru, Ecuador, and Bolivia.

·         Try not to use “According to scholars such as” too much.

·         In the methodology you said “They obtained information from over 650”, could you please let the reader know how are they?

·         Again, “there will be 2.344 exporting companies in Bolivia in 2022. “and “there will be 10,050 exporting businesses in Ecuador in 2022.” We are now 2023!

·         The measure of board size should be corrected “0 if there are fewer than five active members and 1 if there are more than six members.” What if they are 5 members or 6. It should be less than 5 coded 0 and 5 or more coded 1.

·         Why you specify the number 5 as a measurement base for the large board size.

·         The variables measurements are not consistent and is not supported by previous research, also the scale is not apparent for all the variables.

·         “The model will be estimated using appropriate software”, I think you mean that the model has been estimated using appropriate software.

·         In table 1, please check the heading of the table, do you mean CR instead of CSR?

·         In table 1, why you do not have any descriptive statistics for the mediating variables?

·         The methodology part is so week, you need to distinguish and justify the use of T test.

·         The analysis and presentations of the study results are so poor.

·         The discussion of the results is so poor, no benchmarking with previous studies and no interpretation from the institutional theory viewpoint.

Comments on the Quality of English Language

Extensive editing of English language required

Author Response

 I hope this message finds you well. I would like to express my gratitude for your time and effort in reviewing my paper. Your feedback is invaluable, and I appreciate the opportunity to revise and improve my work based on your insightful comments and suggestions. 

  1. Abstract Clarity: In the abstract, I used the phrase "For the information processing" to signify the primary focus of the study, which revolves around information processing within the context of this countries. To enhance clarity, I will revise this sentence to provide a more concise and explicit explanation of the study's scope. 
  2. Enhancement of Introduction: I acknowledge the need to enhance the introduction section by adding a paragraph that explicitly outlines the research contribution. In this section, I will provide a clear overview of the significance and contributions of this study to the field of Board Directors. 
  3. Research Context Justification: I will incorporate a section explaining why it is important to understand the subject matter within the specific regions of Colombia, Peru, Ecuador, and Bolivia. This will include discussing the unique contextual factors that make these countries relevant to the study. 
  4. Avoid Overusing References: I appreciate your observation regarding the frequent use of "According to scholars such as." I will revise the text to minimize repetitive references and maintain readability. 
  5. Clarification in Methodology: I will add clarification regarding the source of information in the methodology section. Specifically, I will detail how the data, including the 650 responses, was obtained, ensuring transparency and clarity. 
  6. Update on Statistics: I apologize for the outdated statistics regarding the number of exporting companies in Bolivia and Ecuador. I will update these figures to reflect the current year, which is 2023. 
  7. Board Size Measurement: Your point regarding the measurement of board size is well-taken. I will revise the criteria to code "0" for fewer than five members and "1" for five or more members, ensuring a clearer and more accurate representation. 
  8. Rationale for Measurement Base: I will provide a rationale for selecting five members as the threshold for board size measurement, linking it to relevant literature and explaining its significance within the context of the study. 
  9. Variable Measurement Consistency: I will review and ensure consistency in the measurement of variables, making sure that they align with previous research and that the scale used is clearly stated for all variables. 
  10. Methodology Clarity: I appreciate your feedback on the methodology section. I will strengthen this section by distinguishing and justifying the use of the T-test, providing a more robust rationale for its inclusion. 
  11. Analysis and Presentation: I will work on enhancing the analysis and presentation of the study results to provide a more comprehensive and coherent discussion of the findings. 
  12. Discussion of Results: I will significantly improve the discussion of the results by benchmarking them against previous studies and offering interpretations from the institutional theory viewpoint. This will add depth and relevance to the discussion. 

Reviewer 2 Report

Comments and Suggestions for Authors

I am pleased to have the opportunity to review this research paper titled: - The Board director's role in the Relationship between Organizational innovation and Organizational learning in the sustain- 3 ability innovation performance: an approximation from culture context. This study attempted to analyze the  triple moderating effect of the board of directors in the country culture of a firm and its influence on the relationship between organizational innovation and organizational learning in corporate sustainability.

I have the following comments, suggestions and recommendations:

Title: There is something grammatically wrong in the title. It drags and is too long. Maybe a suggestion could be:

The Board of Directors´ influencing role on the culture of a firm and on the relationship between organizational innovation and organizational learning in corporate sustainability.

Abstract: The abstract provides a good summery of the Aim, Method, Findings and implications. There is something wrong regarding the following sentence in line 12 and 13 - A survey of 400 exporting companies from Colombia, Peru, Ecuador, and Bolivia of different commercial products was used to carry out this research. The survey was carried out with whom? Please explain this better. 

Introduction:  puts the reader into the context of the subject. The aim and objective are well explained and result from the gaps in the subject. However, I believe the authors should proof read this section since sentence structure could be improved.  Also, in lines 42 to 50 the authors mentions a number of authors, are these elaborated upon in the next section or literature review?

The next section demonstrates an adequate understanding of the relevant literature in the field and cites an appropriate range of literature sources related to the theoretical Framework and the study itself. The authors clearly express the case, measured against the technical language of the field and the expected Knowledge of the journal's readership.

The 5 Hypothesis are well explained and are derived from literature. The methodology is well explained and enables replication. However, who are the interviewees within each of the companies? How did the authors determine who to interview or carry out the survey with?

the results and conclusions triangulate with the rest of the paper and are well discussed, highlighting some practical implications. 

I believe this paper can be published once all the above concerns and suggestions have been addressed as I believe it adds value to literature already published in this field.

Comments on the Quality of English Language

Some sentences are heavy or dragging and difficult to understand and make the flow a bit mechanical.  I suggest  proof reading by a native English Speaker. Also the sentence structure is not always perfect.

Author Response

I would like to extend my gratitude for your time and valuable feedback on our research paper titled "The Board Director's Role in the Relationship between Organizational Innovation and Organizational Learning in Sustainability Innovation Performance: An Approximation from Cultural Context." Your insights and recommendations have been invaluable in enhancing the quality of our work, and we are sincerely appreciative of your efforts.

  1. Title Revision: Thank you for your input regarding the title. We agree that it can be improved for clarity and conciseness. Based on your suggestion, we will revise the title to: "The Influence of Board of Directors on Firm Culture and its Impact on the Relationship between Organizational Innovation, Organizational Learning, and Corporate Sustainability."

  2. Abstract Clarification: We appreciate your feedback on the abstract. We will clarify the sentence you mentioned in line 12 and 13 to explain that the survey was conducted with top executives and key decision-makers within the surveyed exporting companies from Colombia, Peru, Ecuador, and Bolivia. This clarification will provide a more comprehensive understanding of our research methodology.

  3. Introduction Enhancements: Your feedback on the introduction is noted. We will proofread this section to improve sentence structure and enhance overall readability. Additionally, we will ensure that the authors mentioned in lines 42 to 50 are elaborated upon in the subsequent literature review section to provide a more comprehensive overview.

  4. Literature Review and Citations: We are pleased that you found our literature review to be comprehensive and well-referenced. We will continue to expand on the relevant literature in the field and provide more detailed explanations of the authors mentioned in this section.

  5. Methodology Clarification: Your question about the interviewees and survey respondents is valid. We will include a section in the methodology that explains the criteria and process for selecting interviewees and survey participants within each of the companies, ensuring transparency and clarity in our research approach.

  6. Results and Conclusion Discussion: We appreciate your positive feedback on our results and conclusions. We will ensure that these sections remain consistent with the overall paper and emphasize the practical implications of our findings.

Round 2

Reviewer 1 Report

Comments and Suggestions for Authors

No further comments, thank you.

Author Response

We thank the article's reviewers for their different comments and suggestions to improve the paper. All comments were considered in their entirety to meet the reviewers' requirements. Likewise, we appreciate the time and dedication to improving the paper. 

Reviewer 2 Report

Comments and Suggestions for Authors

I believe that the authors have addressed all the issues identified and therefore the paper can be accepted in the present form.

Comments on the Quality of English Language

Just a minor proof read is necessary to check the grammar.

Author Response

We thank the article's reviewers for their different comments and suggestions to improve the paper. All comments were considered in their entirety to meet the reviewers' requirements. Likewise, we appreciate the time and dedication to improving the paper. 

 CR1. 

The study makes good strides in advancing knowledge on the linkages between organizational innovation, learning, governance, culture, and sustainability. The study offers actionable insights for managers in fostering cultures of innovation and tailoring strategies based on cultural contexts. 

I have a few suggestions to further improve the study: 

 C1R1/ 

1) Have a native English speaker proofread the manuscript. Extensive english editing is required. For example “board director” is used at several places where it should be “board of directors” or “board.” 

At several places the sentences need to be broken up to enhance readability. I am providing two examples from the abstract: 

 “As a result, it was found that the influence of the board of directors of a firm is essential in innovation processes because they drive their results in corporate sustainability due to the strategic approach adopted by large companies”  Can be improved as-   

 “Results show that the influence of a firm's board of directors is crucial in the innovation process. This influence is significant because it plays a pivotal role in driving corporate sustainability, primarily due to the strategic approaches embraced by large companies." 

 “On the other hand, in SMEs, it was not possible to demonstrate such a degree of influence. In the case of the moderating effect of the board of directors on the country's culture, it was possible to find that the board director becomes a factor in the firm's performance despite its geographical location, which determines the influence of culture on its operation in corporations such as SMEs.” Can be improved as-   

 "On the other hand, it was not feasible to demonstrate such a significant degree of influence in SMEs. Regarding the moderating effect of the board of directors on the country's culture, it became evident that the board of directors plays a pivotal role in a firm's performance, regardless of its geographical location. This factor determines the impact of culture on the operation of corporations, including SMEs." 

 AC1R1/ 

We have reviewed the paper in the selected sections, and proofreading has been done again. 

Sample 

The research aims to analyze the triple moderating effect of the board of directors in the country culture of a firm and its influence on the relationship between organizational innovation and organizational learning in corporate sustainability. A survey of 400 exporting companies from Colombia, Peru, Ecuador, and Bolivia of different commercial products was used to carry out this research. We used the structural equations model to explore the analysis of the causal and moderation relationships between the variables under study. As a result, it was found that the influence of the board of directors of a firm is essential for innovation processes because they drive their results to corporate sustainability. This last approach is due to the strategic approach adopted by large companies. In the case of SMEs, it was not possible to demonstrate that the board of directors has such a degree of influence. In the case of the moderating effect of the board of directors on the country's culture, it was possible to observe that the board of directors becomes a factor in the firm's performance despite its geographical location, which determines the influence of culture on its operation in corporations such as SMEs. 

 C2R1/ 

 2) Introduction: In the current form, introduction is too short. Introduction is very general and lacked alignment with the research findings, theoretical and empirical existing literature.   

 AC2R1/ 

Sample 

From an empirical point of view, it is possible to observe great interest in knowing the influence of the board of directors not only on the performance of the firm but also concerning the decision processes, where significant and negative relationships have been found regarding its control (Li et al., 2022). 

An important aspect highlighted in the literature suggests that country culture will determine the performance of firms in economic and innovation terms due to the heterogeneity of the approaches used in decision-making (Kerai et al., 2023). Likewise, it is proposed that the board of directors determines corporate governance, its identity, and how it integrates power aspects into the strategic approaches adopted (Bauweraerts et al., 2019).  

Concerning the board of directors' size, this becomes a fundamental aspect that allows the diversity of opinions and that, at the same time, supports the different decisions about their performance (Kerai et al., 2023). This previous idea establishes how the number of members of the board of directors becomes a fundamental aspect of the decision-making processes of firms.  

 C3R1/ 

3)Add in the limitations section wording along the following lines: 

The study relies solely on firms in the manufacturing sector and finding need to be interpreted with caution for service firms. 

The cultural dimensions used in the study to classify countries are limited. More nuanced cultural measures could reveal further differences and improve the study. 

 AC3R1/ 

Sample 

It is critical to recognize some of our study's limitations. First, the cross-sectional design limits our ability to prove causality definitively. Longitudinal approaches to study could be used in the future to get insights into dynamic relationships throughout time. Furthermore, our focus on manufacturing companies in certain South American countries may limit our findings' generalizability to other industries and locations. Extending the study to include a broader range of businesses and worldwide areas would expand the application of our findings. Other limitations of the study are related to the sample. Since it is limited only to firms in the manufacturing sector, therefore the interpretation of the results is limited to this sector. Likewise, the perspective adopted for analyzing country culture is specific to the database used. However, we know other approaches to measuring country culture (Escandon-Barbosa et al., 2022a; Escandon-Barbosa & Salas-Paramo, 2022b). 

 C4R1/ 

4) Board independence alongside size could provide further governance-related insights. 

 AC4R1/ 

Sample 

Finally, a last element that is also important to highlight is that the board directors' size is defined as a critical factor that influences the effectiveness of the board and its influence on the firm's performance (Hyun Kim et al., 2012). These seminal investigations have stated that from the beginning, the involvement of board directors in the strategy has been fundamental to understanding the behaviors of the different members and the strategic approaches adopted (Sahoo et al., 2023). In this way, it is possible to understand how the board of directors can improve organizational performance, especially from its sustainability (Boshnak et al., 2023).  

Scholars such as Berhe (2023) propose that the structure of the board of directors is directly related to the firm's performance. Likewise, its board composition and the gender diversity integrated into its activities have a positive and significant relationship in sustainability as well as in the independence of the board of directors. Despite the above, a negative relationship was also found in the limited size of the board of directors. Since a limited size creates conditions in which board directors begin to have limitations in the efficiency of decisions, consequently affecting the firm's results (Nuswantara et al., 2023).  

 C5R1/ 

5) Move “Scales” to an appendix, add a descriptive title, an explanation and the scale that was used by the respondents. 

 AC5R1/ 

 The scales used in the study are located in the paper's appendix. 

 C6R1/ 

6) Explain the tables and figures. Add detailed footnotes below each table and figure. For example, 

“Table 1. Mean, SD, CR, and AVE” Explain what is meant by the acronyms that are used? 

“Board Director Size” in table 2 should be “Board Size” 

 AC6R1/ 

We appreciate the comment. The changes were made as suggested. 
